# Intersection of the Ubiquitin–Proteasome System with Oxidative Stress in Cardiovascular Disease

**DOI:** 10.3390/ijms232012197

**Published:** 2022-10-13

**Authors:** Min Qiu, Jimei Chen, Xiaohong Li, Jian Zhuang

**Affiliations:** 1Guangdong Provincial Key Laboratory of South China Structural Heart Disease, Guangdong Cardiovascular Institute, Guangdong Provincial People’s Hospital, Guangdong Academy of Medical Sciences, Guangzhou 510080, China; 2Department of Cardiac Surgery, Guangdong Cardiovascular Institute, Guangdong Provincial People’s Hospital, Guangdong Academy of Medical Sciences, Guangzhou 510080, China

**Keywords:** cardiovascular diseases, oxidative stress, ubiquitin–proteasome system

## Abstract

Cardiovascular diseases (CVDs) present a major social problem worldwide due to their high incidence and mortality rate. Many pathophysiological mechanisms are involved in CVDs, and oxidative stress plays a vital mediating role in most of these mechanisms. The ubiquitin–proteasome system (UPS) is the main machinery responsible for degrading cytosolic proteins in the repair system, which interacts with the mechanisms regulating endoplasmic reticulum homeostasis. Recent evidence also points to the role of UPS dysfunction in the development of CVDs. The UPS has been associated with oxidative stress and regulates reduction–oxidation homeostasis. However, the mechanisms underlying UPS-mediated oxidative stress’s contribution to CVDs are unclear, especially the role of these interactions at different disease stages. This review highlights the recent research progress on the roles of the UPS and oxidative stress, individually and in combination, in CVDs, focusing on the pathophysiology of key CVDs, including atherosclerosis, ischemia–reperfusion injury, cardiomyopathy, and heart failure. This synthesis provides new insight for continued research on the UPS–oxidative stress interaction, in turn suggesting novel targets for the treatment and prevention of CVDs.

## 1. Introduction

Cardiovascular diseases (CVDs) represent significant public health and social challenges, with the number of cases estimated at 18 million globally, which is predicted to increase to 23 million [1,2]. CVDs consist of diverse pathologies with numerous origins and manifestations related to the critical cardiac function of continuously delivering blood throughout the body for maintaining life as well as to its complex structure [3]. In general, CVD is a broad disease class comprising stroke, hypertensive heart disease, rheumatic heart disease, peripheral arterial disease, and other vascular diseases, such as atherosclerosis and coronary artery disease, as well as other cardiac pathologies, including ischemia–reperfusion injury and cardiac-remodeling-related defects [4,5,6]. Given the high incidence, morbidity, and mortality, more efforts to improve the survival rate of patients and lower the socioeconomic disease burden are needed, which require a more in-depth understanding of the underlying pathological mechanisms at the cellular and molecular levels.

Oxidative stress seemingly plays a major role in CVD events associated with atherosclerosis, ischemia–reperfusion, diabetes, hyperlipidemia, high blood pressure, and smoking exposure [7]. These conditions impose damage to the antioxidant defense system, which may trigger the excessive production of reactive oxygen species (ROS) and reactive nitrogen species (RNS) [8]. Accordingly, therapeutic strategies that can maintain the balance between oxidants and antioxidants via ROS/RNS scavenging or detoxification to counteract oxidative stress damage could hold promise for preventing and treating CVDs [9]. The main three lines of antioxidant defense include antioxidant molecules, such as antioxidant enzymes, and the proteolytic pathways and proteins involved in oxidative damage repair [10]. The ubiquitin–proteasome system (UPS) is the primary system responsible for the degradation of cytosolic proteins in the repair system [11]. Accumulating evidence indicates that UPS dysfunction is also involved in the development of CVDs [12,13]. Cardiac proteins are in a dynamic state of continual degradation and synthesis; they are entirely replaced within 30 days under physiological conditions [14]. Thus, the UPS regulates this turnover and further plays a role in the cellular response to oxidative stress; consequently, the reduction–oxidation (redox) state may fine-tune the activity of the UPS under certain conditions (Figure 1) [15].

Accordingly, understanding the interaction and mechanisms underlying the link between UPS and oxidative stress could inform new treatment strategies and improve outcomes for patients with CVDs. This review highlights the emerging evidence of the interlinked roles of oxidative stress and the UPS as key mediators driving the CVD process. We particularly focused on key CVDs and related pathophysiological processes, including atherosclerosis, ischemia–reperfusion injury, cardiomyopathy, and heart failure (HF).

## 2. UPS and Oxidative Stress

### 2.1. UPS

Ubiquitin (Ub), first isolated in the 1970s [16], reportedly plays a key role in many cellular processes, including DNA damage repair and proteolysis. Ub is a highly conserved 76-amino acid protein, which acts as a post-translational protein modifier in all eukaryotes [17]. Covalent attachment of the C-terminal glycine residue of Ub to the lysine residues of substrate proteins (mainly damaged, misfolded, or other worn-out proteins)—a process known as ubiquitination—forms a molecular mark for target proteins or downstream regulatory interactions [18]. Although Ub harbors seven lysine residues, the Lys 48- or Lys 11-linked poly-Ub chains are the most common forms of ubiquitination to modify Ub protein substrates, producing the signal for proteasome degradation [19]. Ubiquitination is catalyzed by a three-enzyme cascade involving (1) E1 Ub-activating enzyme, (2) E2 Ub-conjugating enzyme, and (3) E3 Ub ligase (hereafter simply referred to as E3) [20]. It is generally accepted that activated Ub is gradually transferred to its target protein: Ub attaches to a cysteine residue of E1 Ub-activating enzyme in an ATP-dependent reaction, and the activated Ub transfers to a cysteine residue of E2 Ub-conjugating enzyme, which in turn transfers the Ub to a lysine residue of the substrate protein via an E3 ligase. E3 enzymes, as the critical components in the ubiquitination cascade, can be classified into three main groups according to the presence of the (1) Really Interesting New Gene-finger (RING) domain (comprising the largest group, which directly catalyzes the transfer of Ub), (2) homologous to the E6AP carboxyl terminus (HECT) domain (which accepts Ub from E2 Ub-conjugating enzyme), and (3) RING-in-Between-RING (RBR) E3 ligases (with the common features of RING and HECT E3 groups) [21]. The structures of these domains are summarized schematically in Figure 2 [22]. Briefly, RING E3s comprise RING with a zinc-binding domain or U-box domain, which has the same RING fold but without the zinc domain; the HECT E3 domain has a bi-lobar architecture in which the *N*-terminal lobe tethers with the C-terminal lobe via a flexible hinge; RING1 and RING2 domains are separated by an in-between-RING domain (Figure 2 dashed box). E3s can be regulated by neddylation, phosphorylation, and other interactions with proteins or small molecules.

Ubiquitinated proteins are degraded after reaching the 26S proteasome (Figure 2) [23]. With two terminal α-subunit rings and two middle β-subunit rings, the 20S proteasome forms a core complex that combines with the 19S proteosome, a two-cap-like regulated structure consisting of the ATP-dependent 26S proteasome [21]. The polyubiquitin proteolytic signal allows the 26S proteasome to recognize a large variety of protein substrates, which can realize the function of the UPS [24]. Protein ubiquitination is a dynamic and reversible process, catalyzed by a series of deubiquitylating enzymes (DUBs); Ub removed from the substrates can then be sent to an available pool for recycling [19]. Therefore, manipulating protein ubiquitination offers considerable prospects for blocking the pathways contributing to relevant diseases.

### 2.2. Oxidative Stress

Oxidative stress is exerted by the excessive production of ROS/RNS, resulting in an imbalance between oxidant and antioxidant processes [25]. ROS include free radicals such as superoxide (O2•−), hydroxyl (HO•), and peroxynitrite (ONOO•−), as well as non-radical molecules, including hydrogen peroxide (H_2_O_2_), which are mostly derived from aerobic metabolism via oxygen reduction in the mitochondria [26], endoplasmic reticulum (ER) [27], and peroxisomes [28]. A series of enzymes, including NADPH oxidase, endothelial nitric oxide synthase, xanthine oxidase, uncoupled arachidonic acid, and metabolic enzymes such as cytochrome P450 enzymes, lipoxygenase, and cyclooxygenase, mediate enzyme-catalyzed reactions to continuously produce ROS [29]. ROS are also derived from non-enzymatic reactions such as the mitochondrial respiratory-chain-related reaction.

The contribution of an increase in ROS production and their accumulation to CVD events is widely accepted. The main mechanisms underlying this association include an increase in nitric oxide (NO) levels, which induces cardiac dysfunction, ROS-induced (via NADH/NADPH oxidase or cytokine) cardiac apoptosis or necrosis and superoxide-mediated endothelial dysfunction [7].

### 2.3. Oxidative Stress Directly Affects the UPS

The direct effect of oxidative stress on UPS has become widely accepted in recent years [30]. There is no doubt that oxidative stress, related to the production of ROS, is a major upstream component in the signaling cascade promoting cell proliferation, adhesion molecule production, and inflammatory responses [31]. An increase in protein carbonylation occurs during oxidative stress activation, specifically on the 19S regulatory S6 subunit ATPase, considered an oxidation-sensitive protein. The increased oxidative modification of S6 ATPase was observed in H_2_O_2_-treated cells [31]. As a superoxide generator, paraquat reportedly induces the accumulation of ubiquitinated proteins [32]. Indeed, the 26S proteasome reversibly disassembles to the 20S core and 19S regulatory structure upon exposure to moderate oxidative stress [33].

Kelch-like ECH-associated protein 1 (Keap1) was reported to act as an E3 ligase of the transcription factor nuclear factor (erythroid-derived 2)-like 2 (Nrf2), which is essential for the protection of cells against oxidative stress, and its function is modified by ROS [34]. Without the binding of Keap1, Nrf2 is not modified by ubiquitin. It thus remains in a stable state, resulting in increased levels of Nrf2 targets, such as NAD(P)H quinone dehydrogenase (NQO-1), glutathione S-transferase (GST), superoxide dismutase 1 (SOD1), and hemoxygenase-1 (HO-1), which play a role in the repair and response to oxidative injury [35]. Moreover, ROS activate redox-sensitive proteolytic ligases (such as MuRF1 and atrogin-1) of the UPS [36]. The degradation susceptibility of extensively oxidized proteins decreases due to the formation of protein aggregates [37]. Moreover, all of the components of the UPS, including DUBs, can be impaired by extensive or chronic oxidative stress, which induces the dissociation of the 20S core particle from the 19S regulatory particle of the 26S proteasome in eukaryotic cells, leading to impaired 26S proteasome activity and the consequent accumulation of ubiquitinated proteins [38]. Thus, the UPS could be an important target of oxidative stress, although the detailed mechanism remains to be elucidated.

### 2.4. Role of the UPS in Oxidative Stress

The UPS is considered the third line of antioxidant defense, which may alleviate the damage caused by ROS, regulating the oxidative stress status [11]. Oxidative stress predisposes proteins to misfolding and toxic aggregation, which can be prevented by the protein quality control system, including the UPS [39]. Thus, dysfunction of the UPS results in the accumulation of oxidative-stress-damaged proteins and a further increase in ROS, forming a vicious cycle that exacerbates oxidative stress [40]. The UPS also directly affects the degradation of some key enzymes in oxidative stress, including the Nrf2-Keap1 interaction, an important oxidative stress sensor, as described above. When redox processes are in equilibrium, Nrf2 translocates from the nucleus to the cytoplasm, where it is degraded by ubiquitination; thus, a dysfunctional UPS would disrupt the redox balance [41]. Silent mating type information regulation 2 homolog 3 (SIRT3), a mitochondrial deacetylase that can eliminate excessive ROS, was found to be degraded by ubiquitination to induce cell death [42], representing yet another example by which the UPS interacts with oxidative stress, controlling most cellular and disease processes. Indeed, proteasome activity may be increased under low-level oxidative stress but may be inhibited under high-level oxidative stress. In addition, the anti-oxidative ability of endothelial cells can be enhanced by low-level proteasome inhibition, whereas oxidative stress may be initiated under high-level proteasome inhibition [43].

## 3. Interaction of the UPS and Oxidative Stress in CVDs

### 3.1. Atherosclerosis

Atherosclerosis is a complex progressive disease affecting larger- and medium-sized arteries, contributing to many types of CVDs, including myocardial infarction and ischemic HF [44]. A consensus has been reached that atherosclerosis consists of three main phases: (1) a long asymptomatic initial stage with fatty streaks; (2) a progression stage characterized by the formation of atheromatous plaques over several decades; and (3) plaques-related complications, including plaque erosion or rupture, thrombus formation, and complete or near-complete vascular occlusion [43]. Atherosclerosis mainly results from pathophysiology linked to oxidative stress, lipid metabolism alterations, and inflammation, which are associated with several molecules such as NO, adhesion molecules, and nuclear factor-kappa B (NF-κB). Accumulating evidence also points to an important role of UPS in all three stages of atherosclerosis [43].

The UPS regulates Nrf2 and hypoxia-inducible factor 1 (HIF-1) as the major regulators of oxygen homeostasis [38]. HO-1 and SOD1, as the targets of Nrf2, were shown to have anti-atherogenic protection effects, which are also regulated by the UPS [38,45]. Binding to antioxidant response elements in the nucleus promotes the transcription of antioxidant genes, including Nrf2/Keap1, as one of the most powerful intracellular antioxidative pathways. In addition to atherosclerosis, many other CVDs (described in detail below) are also promoted by the dysfunction of Nrf2/Keap1 [41]. Under physiological conditions, as an E3 ubiquitin ligase, Hippel–Lindau tumor suppressor protein (pVHL) ubiquitinates hydroxylated HIF-1α [46], which generates a state of low HIF-1α abundance after its rapid proteasomal degradation [47]. However, this process is inhibited under hypoxic conditions, in which HIF-1α translocates to the nucleus and combines with HIF-1β to ultimately promote the adaptation of cells to hypoxia. Notably, ROS and some pro-inflammatory factors, such as NF-κB, can also activate HIF-1α [38]. Two of the NF-κB family members, p50 and p65, are normally distributed in the cytosol of healthy vessels, whereas the complex remains in the nucleus in the atherosclerosis disease state [48]. Interestingly, the proteasome was shown to regulate nuclear NF-κB signaling through the degradation of the NF-κB inhibitor (IκBα) in the canonical activation pathway [49]. The proteasome also mediates the removal of p50/p65 via its proteolytic activity in the non-canonical pathway, which is activated by oxidative stress [50,51].

The endothelial injury initiated by oxidative stress contributes to the development of atherosclerosis and other CVDs. A recent study confirmed that WWP2, a HECT-type E3 ubiquitin ligase, interacts with Septin4, a known endothelial injury factor [52], to promote Septin4 degradation via the UPS, thereby preventing endothelial injury and vascular remodeling. This counteraction was disturbed by the decreased expression of WWP2 under oxidative stress and angiotensin II-induced endothelial injury [53]. Aggregated low-density lipoproteins (LDLs) reportedly induced the ovine E2 ubiquitin-conjugating enzyme E2–25K, followed by the promotion of p53 degradation, which may suppress the apoptosis of foam cells [54]. Interestingly, oxidized LDL (Ox-LDL) was found to induce the apoptosis of macrophages via inhibiting proteasomal activity [55]. Ox-LDL also induces endothelial dysfunction to cause atherosclerosis through the increased ubiquitination of sirtuin 1 (SIRT1), an anti-inflammatory and anti-autophagy cytokine, under the regulation of cytoplasmic polyadenylation element-binding protein 1 (CPEB1) that, in turn, reduces the expression of lectin-like Ox-LDL receptor-1 (LOX-1), consequently regulating ROS production [56]. In addition, Ox-LDL levels were elevated in a patient with diabetes due to the production of mitochondrial ROS promoted by intracellular hyperglycemia, which ultimately intensified atherosclerosis [57].

Increasing evidence indicates that the major risk factors for atherosclerosis, including diabetes, obesity, hypertension, and aging, are closely related to oxidative stress. One study emphasized that increased oxidative stress decreased the activity of the UPS in the context of atherosclerosis progression [37]. Moreover, the role of deubiquitination is particularly relevant in many of these processes. As a dynamic organ secreting adipocytokines and affecting vascular function, the adipose tissue that accumulates in obesity transforms the vascular-protective or anti-atherogenic environment to a pro-atherogenic phenotype [58]. For instance, wingless-related integration site 5A (WNT5A) could induce oxidative stress and interact with the deubiquitinating enzyme ubiquitin-specific protease 17 (USP17) to play a key role in activating the redox-sensitive migration of vascular smooth muscle cells [59]. In addition, UPS inhibitors were reported to reduce oxidative stress and alleviate atherosclerosis [38]. These conflicting observations highlight the complicated mechanisms underlying the contributions of UPS and oxidative stress to atherosclerosis (Figure 3).

### 3.2. Ischemia–Reperfusion Injury

The erosion and rupture of coronary atheromatous plaques and vascular occlusion result in severe myocardial infarction and ischemia, with the pathology characterized by impaired function of the myocardium and enhanced cardiomyocyte apoptosis [60]. This combination prevents the possibility of repair and regeneration of the damaged myocardium owing to the insufficient proliferation capacity of cardiomyocytes [61]. Although timely reperfusion following an injury can delay the negative outcomes of ischemia and suppress cell death, the inevitable damage to both cardiomyocytes and the myocardium caused by reperfusion itself, known as ischemia–reperfusion injury (IRI), also results in hypoxia and microvascular or myocardial dysfunction [62]. Ischemia–reperfusion injury is inevitable in the course of several surgical procedures, including organ transplantation, cardiothoracic surgery, and general surgery, due to an imbalanced metabolism [61]. A dysfunctional microvasculature, cell death, activated leukocytes, and accumulated chemokines and ROS are the main pathophysiologic mechanisms of IRI [63].

Indeed, ROS and cardiac inflammation form a vicious cycle, in which ROS initiate a cascade reaction of phosphorylation/dephosphorylation and inflammation, which further fuels ROS production, ultimately promoting IRI [62,63]. Many cytokines, including HIF-1, tumor necrosis factor (TNF), interleukins (ILs), and NF-κB, are involved in the inflammatory reactions in response to oxidative stress in ischemia [64]. Moreover, the levels of several antioxidant factors, such as GST and SOD, which eliminate ROS under physiological conditions, were reportedly decreased in rat ischemic hearts and other organs. Their reduction was also found under reperfusion in contrast to expectations [65,66]. Oxidative stress thus provides a strong foundation for IRI progression.

Considering the relationship among IRI, oxidative stress, and cardiac inflammation, it is rational to focus on the function of the UPS in this mechanism, considering its critical role in the regulation of inflammatory processes and the degradation of misfolded or damaged proteins. Accordingly, substantial research focusing on IRI therapy in the last two decades has examined whether inhibition of the UPS can alleviate the associated inflammation or oxidative stress, even the injury itself, demonstrating promising results for clinical applications.

Inhibition of the NLRP3 inflammasome was shown to ameliorate adverse cardiac remodeling and decrease the infarct size in an IRI animal model [67], whereas excessive ROS activated the NLRP3 inflammasome [68]. Once the Keap1-Nrf2 interaction was blocked, the Nrf2 signaling pathway was activated and the NLRP3 inflammasome was inhibited, subsequently leading to a reduction in IRI [69]. LDL receptor-related protein 6 (LRP6), a Wnt co-receptor [70], interferes with the interaction between HSF1 and glycogen synthase kinase 3β (GSK3β), a Ser/Thr protein kinase inactivated by phosphorylation, resulting in the inhibition of HSF-1 ubiquitylation under oxidative stress following myocardial ischemia–reperfusion by reducing apoptosis and promoting nucleus stabilization [71].

Increasing research has demonstrated that several UPS inhibitors, including MG132, epoxomicin, bortezomib, PR-39, and PR-11, have a protective effect on the heart during IRI [72], providing strong evidence of the role of UPS on the pathological process. Notably, the activation of IRI has also been shown to result in UPS dysfunction [73]. Blocking of the degradation of IκB, an NF-κB inhibitor, by the proteasome inhibitors PR-39 and PR-11 caused the translocation of NF-κB to the nucleus, which ultimately prevented the release of inflammatory factors in an IRI rat model [74]. Moreover, as ROS can directly activate the NF-κB pathway and thus promote the production of inflammatory cytokines [75], antioxidant therapy and alleviating cardiac inflammation are considered effective strategies to lessen IRI. Direct blockage of the UPS has also been explored as a potentially effective therapy direction for IRI given the effects on alleviating inflammation and oxidative stress [72], although further investigation is needed to establish the safety of this treatment.

### 3.3. Cardiomyopathy

Cardiomyopathies can be divided into primary (genetic, acquired, or mixed) and secondary categories, including hypertrophic cardiomyopathy (HCM), dilated cardiomyopathy (DCM), and restrictive cardiomyopathy (RCM). Cardiomyopathies can also be classified according to their secondary causes, including ischemic, metabolic, auto-immunogenic, infectious, toxic, and neuromuscular [76]. UPS dysfunction has been widely suggested as a key pathogenic factor contributing to the development of cardiomyopathies [77], with ROS production regarded as the cause underpinning this impairment.

Despite this clear link, the interaction of ROS and the UPS has not been a focus of the majority of research on cardiomyopathies, although many studies have focused on one of these factors (ROS or UPS) or their combination with other mechanisms (such as apoptosis or ferroptosis) [76,78,79]. The first examination of the proteasome function in human HCM concluded that UPS dysfunction in HCM resulted from mutations in sarcomere proteins [80], including cardiac troponin T (TnT), myosin heavy chain (MHC), and cardiac myosin binding protein C (cMyBP-C) [77], which are linked to more than half of the cases of familial HCM. The research suggested that ATP could promote the assembly of a damaged 19S proteasome to the 20S core [80] and that oxidation may be an important mechanism involved in UPS dysfunction in human cardiomyopathy or HF. Higher levels of ubiquitinated and oxidized proteins, which may be related to a reduction in proteasome activity, were found in the heart of a murine model with a TnT mutation. In addition, mutated cells exhibited an accelerated energy-produced pattern, which may lead to increased ROS. These oxidated proteins damaged the proteasome subunits and were degraded [81].

Proteasome levels and oxidative stress also appear to be increased in patients with DCM, and the oxidative stress-induced increased 26S proteasome levels might be a compensatory mechanism [82]. UPS proteins co-localizing with oxidation-induced modifier molecules in cardiomyocytes were also suggested to be related to the pathophysiology of DCM. Another mutation of TnT was suggested as a causal mutation of HCM, which was associated with an impaired proteasome, increased ATP production, and elevated levels of stress-related proteins in mice [83]. ITCH (a ubiquitin E3 ligase Itchy homolog) controls the ubiquitin–proteasome degradation of thioredoxin-interacting protein and ameliorated ROS-induced cardiotoxicity in a doxorubicin-induced cardiomyopathy model, ultimately attenuating cardiac hypertrophy [84]. Conditional cardiac-specific HUWE1 (a ubiquitin E3 ligase) knockout mice developed cardiac hypertrophy, accompanied by impaired mitochondrial energy metabolism and ROS defense [85]. The reversal of murine double minute 2 (MDM2), a p53-specific E3 ubiquitin ligase, and SIRT1 reduced the activity of the apoptosis factor p53 through SIRT1-mediated p53 deacetylation and MDM2-mediated p53 ubiquitination, which alleviated oxidative stress and cell apoptosis in the tissues and cells of a cardiomyopathy model induced by adriamycin [86].

The mutation of folliculin-interacting protein 1 (FNIP1) was also reported as a causal mutation of HCM [87]. A recent study [88] demonstrated that CUL2FEM1B (a ubiquitin E3 ligase) targets reduced FNIP1, which alleviates the reductive stress caused by excessive antioxidative processes and could supplement the physiological ROS [89]. Mitochondrial ROS production is regulated by degraded or stabilized FNIP1 to maintain the balance of redox reactions [90]. The transcription factor forkhead box protein O 1 (FoxO1) is related to cardiac hypertrophy, and SIRT3, activated by oxidative stress, can combine with FoxO1 to activate its deacetylation, promoting the expression of the downstream molecules Muscle-RING-finger-1 (MuRF1) and Muscle-Atrophy-F-box (MAFbx/atrogin-1, an E3 ubiquitin ligase), ultimately alleviating myocardial hypertrophy [91]. Interestingly, another study found that LncDACH1 facilitated the degradation of SIRT3 by ubiquitination, promoting mitochondrial oxidative injury and cell apoptosis in the heart of a diabetic mouse model [79]. In addition, by promoting phosphatase and tensin homolog (PTEN), proteasomal degradation was found to be an effective therapy to reduce oxidative stress and injury by disrupting downstream FoxO3 function in diabetic cardiomyopathy [92].

As mentioned above, proteasome inhibitors can alleviate the IRI, and a study in healthy pigs further showed that proteasome inhibition caused cardiac dysfunction; indeed, patients using a proteasome inhibitor for cancer therapy have a higher incidence of HF [77]. Although many studies failed to elucidate the detailed link between UPS and oxidative stress, owing to the complexity of these processes, further study is warranted in this regard to best exploit the therapeutic potential.

### 3.4. Heart Failure

HF is a progressive disease and the final stage of CVD, involving myocardial cell loss, IRI, and the structural remodeling of the myocardium (including cardiomyopathy); thus, the UPS is expected to play a pathophysiological role in HF [93]. Moreover, ROS production has been revealed as a key factor in HF development [94].

Recently, many studies have revealed the mechanisms underlying the interaction between the UPS and oxidative stress, with some showing that the UPS plays a bidirectional role in response to oxidative stress. The E3 ligase tripartite motif-containing protein 16 (TRIM16) was demonstrated to act as a suppressor of pathological cardiac hypertrophy (relieve peroxiredoxin 1 (PRDX1) phosphorylation and oxidative stress) and indicated that targeting the TRIM16–PRDX1 axis is a promising therapeutic strategy for hypertrophy-related HF [95]. However, the ubiquitin E3 ligase TRIM21 suppressed the p62-Keap1-Nrf2 antioxidant pathway in a doxorubicin-induced cardiac dysfunction model, and *Trim21* knockout mice were protected from HF [96]. P21-activated kinase 2 (PAK2) targets Nrf2/Keap1 ubiquitination by mediating 3-hydroxy-3-methylglutaryl reductase degradation 1 (HRD1) expression, thereby alleviating detrimental ER stress and HF, offering another potential therapeutic strategy for HF [97]. Moreover, in a rat model of chronic HF, miR-129-5p was found to target Smad ubiquitin regulatory factor 1 (SMURF1) and repress the ubiquitination of PTEN to improve cardiac function [98]. MiR-454 impaired the expression of neural precursor cells, developmentally downregulated the 4-2 (NEDD4-2)/tropomyosin receptor kinase A (TRKA)/cAMP axis in cardiomyocytes injured by oxidative stress, and its expression was found to be downregulated in a rat HF model [99]. Furthermore, aerobic exercise training reportedly reduces poly-ubiquitinated protein levels in the failing heart, although the mechanism remains unknown [100,101].

Oxidative stress can also activate ubiquitination to contribute to HF pathology. The level of La ribonucleoprotein domain family member 7 (LARP7), a regulator of the DNA damage response linked to ROS, was found to be reduced in the heart under the conditions of HF; accumulated ROS promoted LARP7 ubiquitination and degradation, and reduced sirtuin1 (SIRT1) stability and deacetylase activity, ultimately impairing oxidative phosphorylation and cardiac function [102]. In addition, pVHL expression was upregulated in a DCM mouse model, and oxidative stress induced phospholamban degradation in an in vitro HF model [103]. Moreover, sodium sulfide was found to attenuate ischemic-induced HF by enhancing UPS function in an Nrf2-dependent manner [104].

Collectively, these studies highlight a promising prospect for UPS–oxidative-stress-derived therapies. As an end stage of heart disease, all of these mechanisms associated with other CVDs could be valuable directions in HF research.

## 4. Conclusions

CVDs continue to represent major threats to human life, and the UPS is emerging as a key regulatory pathway in CVD pathology, especially with respect to its role in the regulation of and response to oxidative stress. Cardiomyocyte death and myocardial remodeling are the primary pathological hallmarks of CVDs, which are accelerated by impairment in the cellular redox process and UPS. The UPS appears to play different roles in the different stages of CVDs, contributing to disease development in some cases and improving oxidative stress to delay disease progression in others. Thus, it is important to contextualize the interplay between the UPS and the redox process. Although therapeutic targets for preventing or delaying the progression of CVDs are currently scarce, further research on the UPS–oxidative stress interaction shows promise in this regard.

## Figures and Tables

**Figure 1 ijms-23-12197-f001:**
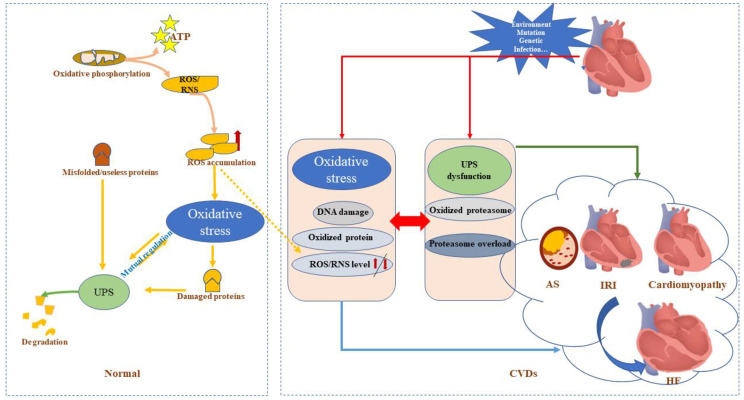
General schematic illustration of the relation among the ubiquitin–proteosome system (UPS), oxidative stress, and cardiovascular diseases (CVDs). **Left**: Oxidative phosphorylation produces ATP and ROS, triggering the UPS to degrade the proteins exposed to oxidative damage along with misfolded proteins; thus, mutual regulation mechanisms are at play between the UPS and oxidative stress. **Right**: The function and/or structure of the heart can be affected by gene mutation, heredity, infection, environmental exposures, and other factors, leading to a series of pathophysiological changes or protein modifications, including oxidative stress and ubiquitination. UPS can degrade the proteins damaged by oxidative stress; however, UPS dysfunction can directly lead to CVDs. Ubiquitination of antioxidant-system-related proteins can further upregulate the levels of reactive oxygen species (ROS)/reactive nitrogen species (RNS) to aggravate oxidative damage. Oxidative stress can directly damage UPS-related proteins leading to abnormal UPS function. Oxidative stress can also overload the UPS, yielding cardiac abnormalities such as atherosclerosis (AS), ischemia–reperfusion injury (IRI), cardiomyopathy, and heart failure (HF).

**Figure 2 ijms-23-12197-f002:**
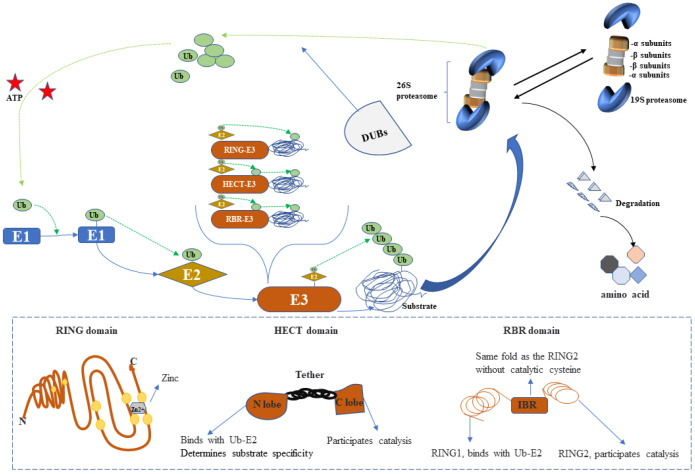
Depiction of the ubiquitin–proteasome system (UPS). The ubiquitin-activating conjugating and ligating enzymes are indicated as E1, E2, and E3. The 26S form of the proteasome catalyzes the degradation of ubiquitylated protein substrates. E3 enzymes are classified into three groups according to the presence of the HECT, RING, and RBR domains. Ubiquitin is transferred from E2 ligases to HECT or RBR-E3 ligases to ultimately ubiquitinate the substrate or directly ubiquitinate the substrate via the catalysis of RING-E3 ligases. Deubiquitination enzymes (DUBs) remove ubiquitin from substrates, which is recycled into cytosolic pools.

**Figure 3 ijms-23-12197-f003:**
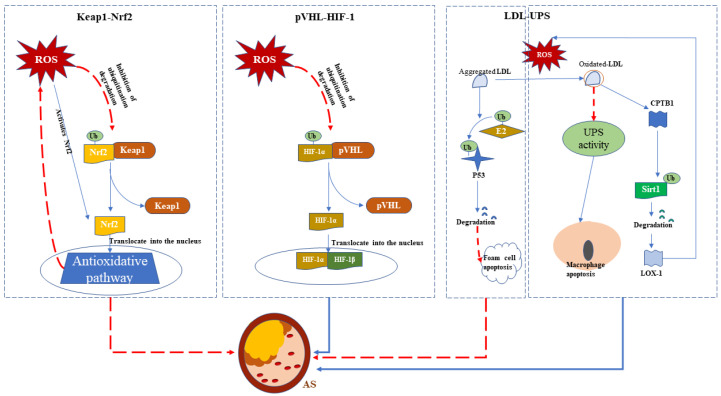
Three main pathways linking the ubiquitin–proteosome system (UPS) and oxidative stress in atherosclerosis (AS) progression. The red dotted line indicates inhibition or alleviation, and the solid blue line indicates promotion or activation. Ub, ubiquitin; ROS, reactive oxygen species; LDL, low-density lipoprotein.

## Data Availability

Not applicable.

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
