# Peer review of "Intersection of the Ubiquitin–Proteasome System with Oxidative Stress in Cardiovascular Disease"

_ijms, 2022, doi:10.3390/ijms232012197_

Round 1

Reviewer 1 Report

While the subject of the work was of high interest at first, my interest as reader and reviewer rapidly decreased due to the poor quality of English. Multiple words were badly selected, giving many sentences an inadequate or imprecise meaning. Accodingly, I strongly recommand to get the text revised by a professional English-native writer.

The introduction set the stage for a review about ROS/RNS in various cardiovascular conditions where the UPS is affected. Instead, the text rather presents the dysfunction of the UPS, and a long list of its components, in various cardiopathies. Sometimes I felt that the authors included a lot of information without much rational justification, instead of focusing on  somes selected examples to illustrate the consequences of dysfunction.

 have noted that some sentences in the text are well written while others have English problems. Somethime this happens in a single sentence such that English is not the same quality before and after a coma. Accordingly, I found some sections of sentences that are identical in the work of other published previsouly, which raise some important concern here.

Reviewer 2 Report

This review is well written and meaningful for general readers to understand the relationships between the UPS and redox regulation.

I suggest the authors improve the manuscript more significantly.

1: Fig1 is hard to understand and elusive.

The authors should illustrate the simple and essential point more impressive.

I think numbering the cascade and unifying each arrowhead.

2: In Fig2, the authors should illustrate the domain structure of three categories of E3s. It should be required by general readers to understand the molecular mechanism and property.

3: I recommend the authors add a figure that explains each E3 and substrate and biological function. This review describes many E3s and proteins that could be difficult to understand the overall stories.

Round 2

Reviewer 1 Report

Great amelioration when compared to the original manuscript. I can only say that this kind of quality should have been proposed the first time the manuscript evaluated.

Line 56. Remove the return to avoid generating a new paragraph.

I acknowledge the work of the authors that greatly enhanced the figures. Now, if possible, increase the resolution of the figures because they appear slightly fusy when reviewing the manuscript.
